# The Heat Transfer Analysis of an Acting-type Heat Retention Panel used in a Hot Rolling Process

**Jiin-Yuh Jang [1,\*], Jin-Wei Guo [1] and Chih-Chung Chang [2]**

[1]  Department of Mechanical Engineering, National Cheng Kung University, Tainan 70101, Taiwan; n16064307@mail.ncku.edu.tw
[2]  China Steel Corporation, Kaohsiung 81233, Taiwan; 165670@mail.csc.com.tw
\*  Correspondence: jangjim@mail.ncku.edu.tw; Tel.: +886-6-208-8573

**Abstract:** In the hot strip rolling process, improvement of the mechanical properties of the strip steel has been a focus for a long time. After rough rolling, high temperature transfer bars are transported by conveyors through the heat retention panel in order to decrease the temperature difference between the head and the tail of the transfer bars. During the heat retention process, the temperature distribution of the transfer bars have a great influence on the mechanical properties of strip steel. A three-dimensional numerical model of a traditional passive heat retention panel is developed to investigate the temperature difference between the head and the tail of the transfer bars. A comparison with the in-situ data from a steel company in Taiwan shows that the present model works well for the prediction of temperature values of the head and the tail and temperature differences of the transfer bars. Based on the developed model, a three-dimensional numerical model of the acting-type heat retention panel is constructed in order to predict whether the temperature difference decreases during the heat retention process.

**Keywords:** hot strip rolling; transfer bar; heat retention panel; heat transfer

## 1. Introduction

The steel industry in the global market is booming. Due to increased market competition, the demand for quality steel products has attracted a high level of attention in recent decades. In order to become more competitive in the steel industry, improving the ironmaking process and increasing the quality control in manufacturing have become a focus of steel plants.

Most steel products are manufactured after high temperature slabs are rolled by rolling mills. In the hot strip rolling process shown in Figure 1, the high temperature slabs are roughly rolled into the transfer bars and then transported into the finishing rolling through the original heat retention panel. During the transport process, in addition to self-radiation heat dissipation, the convective heat dissipation of air, the contact heat conduction cooling of the rollers beneath the conveyors, and the change in the speed of the conveyors will cause a temperature difference between the head and the tail of the transfer bar. When this temperature difference is too large, it will make the finishing rolling more difficult and also cause the production line to shut down.

There are three types of equipment that can improve decreases in the temperature difference in the transfer bars: an inducting heater, an acting-type heat retention panel, and a heat coil box. This study mainly focuses on the acting-type heat retention panel. An acting-type heat retention panel involves installation of a radiation plate inside a traditional passive heat retention panel and a heat source, which can be supplied by burning natural gas or an electric heater, which is used to heat the radiation plate. Eventually, the radiation plate will heat the head and the tail of the transfer bar by radiation heat transfer so that the temperature difference can be decreased. In response to global energy conservation

and carbon reduction, as well as goals of green production, numerous numerical models and methods for the prediction of the hot strip rolling process have been developed and successfully applied to many steel plants but there have been few studies focusing on the heat retention panel.

According to a hot rolling plant layout, Wang et al. [1] designed a space for installing heat retention panels. Bu [2] analyzed the hot rolling assembly line without a heat retention panel and discussed the influence of temperature at the finishing entrance. Furthermore, he analyzed the hot rolling assembly line with an installed heat retention panel. The results indicated that a heat retention panel can effectively decrease the temperature difference between the head and the tail of the transfer bar. Zhang et al. [3] utilized a two-dimensional simplified model to simulate a hot rolling assembly line, both with a heat retention panel and without a heat retention panel. They assumed the heat transfer mode of transfer bar in the heat retention panel to be based on radiation. Thus, they ignored the convective effect of air, and in order to simplify the computational calculation, the influence of the rollers beneath the conveyor was also ignored. Compared with the in-situ data in a steel plant, the results worked well for the prediction of the temperature history of the transfer bar. Zhang et al. [4] utilized a two-dimensional symmetry finite element method (FEM) to simulate the transfer bar inside a heat retention panel with a high radiation coating. They sprayed a high radiation ceramic coating on the inner wall of the panel. Their results showed that a high radiation ceramic coating can significantly decrease the temperature gradient in sections of the transfer bar.

In addition, a significant amount of data on the hot strip rolling process, such as the size of the slab before rolling, the temperature of the slab at the exit of the reheating furnaces, and the running speed of conveyors, has been found to be important parameters to affect the quality of steel products. Bu et al. [5] obtained numerous in-situ data from a steel plant, including the size and properties of slab, the temperature at the exit of the slab reheating furnaces, and the size and the temperature distribution of the transfer bar after rough rolling. They used the Microsoft Visual Basic program to design a computer-aided design program to calculate effect of the temperature of the transfer bar on the finishing rolling force. Ling [6] analyzed the advantages and disadvantages of a traditional passive heat retention panel, an acting-type heat retention panel, and a heat coil box. The results indicated that a traditional passive heat retention panel consumes the least energy but it is inefficient for heat retention. Although an acting-type heat retention panel consumes the most energy, it is the most effective for retaining heat in the transfer bar. A heat coil box exhibits a good performance for heat retention and consumes less energy than an acting-type heat retention panel, but the cost of installation is the most expensive. Speicher et al. [7] used the finite difference method (FVM) combined with heat conduction and heat convection to investigate the impact of the rollers beneath conveyors on slab. Legrand et al. [8] studied the thermal fatigue effect of the rollers beneath the conveyors on slab.

Chielo et al. [9] used the nonlinear heat transfer equation to predict the temperature of the steel on the run-out table (ROT) process. The results show that the performance with a cooling stop temperature concept to the temperature and property of the steel was greatly improved. Mei et al. [10] investigated the strip steel, which was heated by induction heater by the finite element method. They found that the temperature difference became more and more obvious with the increase of thickness. Shulkosky et al. [11] developed a program which allows users to set-up their hot strip mill configuration and simulate the mechanical properties of the steel in the hot rolling process. The program includes reheating furnace, roughing mill stands, heat retention equipment (panels and coil box), finishing mill stands, the run-out table, and the mill exit area. Panjkovic [12] designed a model to predict strip temperature from the roughing mill exit to the finishing mill exit. The results were compared to those from the plant measurements, and it was shown that this model worked very well. Grajcar et al. [13] used a semi-industrial physical model to simulate thermomechanical rolling and controlled cooling of advanced high-strength steels with increased Mn and Al content. The results indicated that the high-quality strip samples with a thickness up to 3.3 mm could be obtained by using heat retention panels. Tudball and Brown [14] developed a transient 3D finite element model to obtain thermal variations during the hot rolling process. The numerical model showed that the temperature

results can provided a relatively accurate prediction with less than 10% deviation. Delpature et al. [15] studied the active tunnel furnace in order to minimize heat losses of the transfer bar on the roller table between roughing mill and finishing mill. With traditional passive heat retention panels, the temperature difference in the head and the tail of the carbon transfer bar was about 20 °C. They found that the active heat retention panels was able to compensate for this drop in temperature.

Based on the studies referenced above, it is important to decrease the temperature difference between the head and the tail of the transfer bar to enhance the quality of steel products. The traditional passive heat retention panel is widely used in many steel plants, but its ability to hold temperature is inefficient. In order to solve this problem, it is necessary to develop other heat retention equipment as soon as possible. Based on the theory of fluid mechanics and the theory of heat transfer, in this study, the commercial software ANSYS-FLUENT combined with UDF (User-Defined Functions) are first used to simulate the traditional passive heat retention panel used in the China Steel Corporation (CSC), Taiwan. After the above results were proved to work well in terms of prediction, an acting-type heat retention panel model was constructed to investigate the performance of both types of heat retention panels.

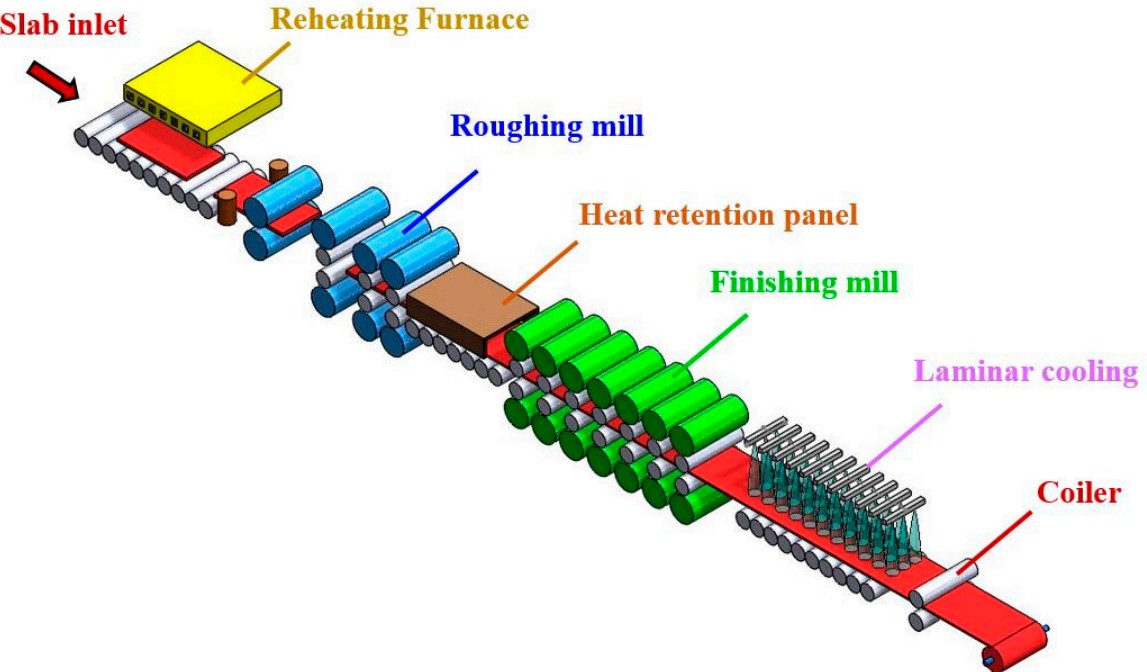

**Figure 1.** Hot strip rolling process configuration.

## 2. Mathematical Analysis

### 2.1. Physical Model

The traditional passive heat retention panel used in CSC has dimensions of 80 m (length) × 3 m (width) × 1.12 m (height). The schematic of the traditional passive heat retention panel is shown in Figure 2a. There are two temperature-measuring devices used to record the temperature of the transfer bars. One is located 3.5 m in front of the heat retention panel and labeled R2DT, and the other one is located 1.5 m behind the heat retention panel and labeled FET. The heat retention panel is divided into two zones, a non-heat retention zone and a heat retention zone. The transfer bars used in the study were stainless steel with dimensions of 70 m (length) × 1.2 m (width) × 0.03 m (thickness). In the process, the transfer bars are transported by the rollers beneath conveyors with various running speeds. The physical model is too large to perform the numerical calculation. Accordingly, the width of the heat retention panel and transfer bar are considered to be a one-half symmetric model in order to reduce the huge computational time required. Relatively, the running speed of the transfer bars were

changed at different time intervals. The length of the transfer bars were assumed to be 85 m, so that it is would be convenient to calculate the location of the transfer bars. As shown in Figure 2b, the simplified three-dimensional heat retention panel has dimensions of 80 m (length) × 1 m (width) × 1.12 m (height) and the transfer bar has dimensions of 85 m (length) × 0.4 m (width) × 0.03 m (thickness). The properties of the heat retention panel, transfer bar, rollers, and heat insulation wool are listed in Table 1.

**Table 1.** Thermal properties of the heat retention panel system.

| Material | $\rho$ (kg/m$^3$) | $C_p$ (J/kgK) | $k$ (W/m K) | Emissivity |
|---|---|---|---|---|
| Transfer bar | 8030 | 550 | 23.8 | 0.75 |
| Heat retention panel | 400 | 1130 | 1.5 | 0.90 |
| Radiation plate | 7940 | 460 | 32.0 | 0.50 |
| Roller | 7700 | 421 | 33.2 | 0.50 |
| Heat insulation wool | 400 | 1313 | 0.3 | 0.10 |

The acting-type heat retention panel involves installing a radiation plate, which has dimension of 80 m (length) × 3 m (width) × 0.02 m (thickness), inside the traditional passive heat retention panel, as shown in Figure 3a. The same as the traditional passive heat retention panel, the width of the acting-type heat retention panel, the transfer bar, and the radiation plate are considered to be a one-half symmetric model in order to reduce the huge computational time required. The length of the transfer bars is also considered to be 85 m. As shown in Figure 3b, the simplified three-dimensional heat retention panel has dimensions of 80 m (length) × 1 m (width) × 1.12 m (height) and the transfer bar has dimensions of 85 m (length) × 0.4 m (width) × 0.03 m (thickness).

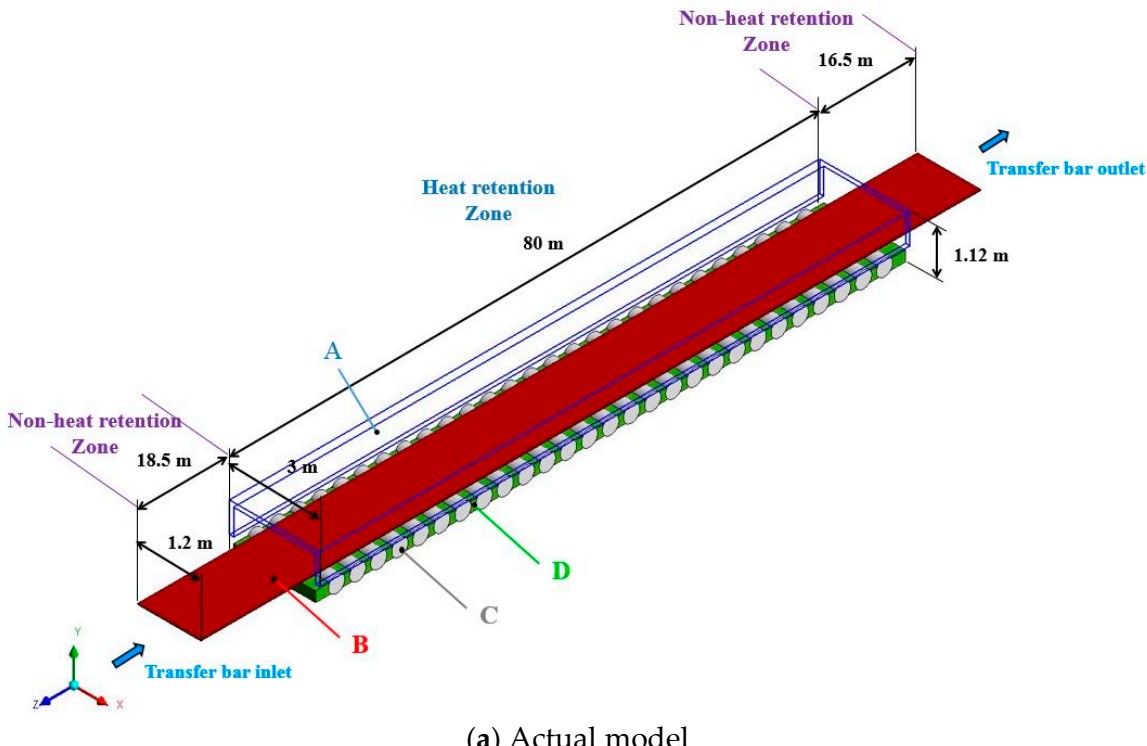

(**a**) Actual model

**Figure 2.** *Cont.*

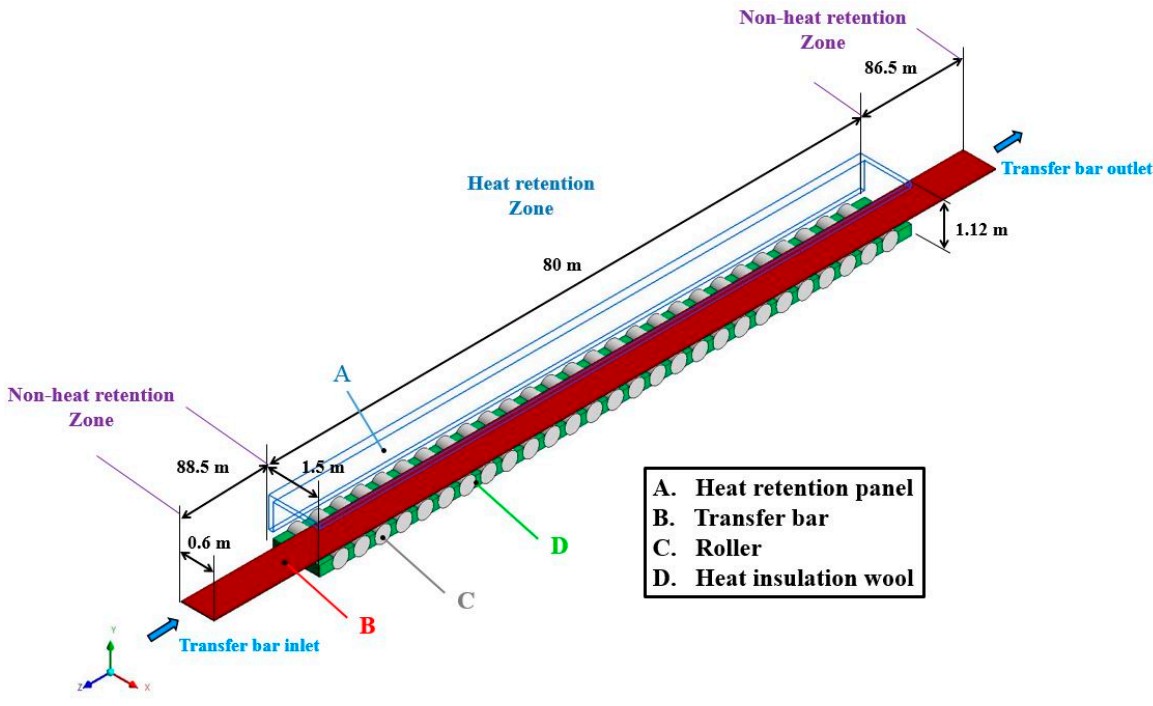

(**b**) Simplified model

**Figure 2.** Physical models of the traditional passive heat retention panel.

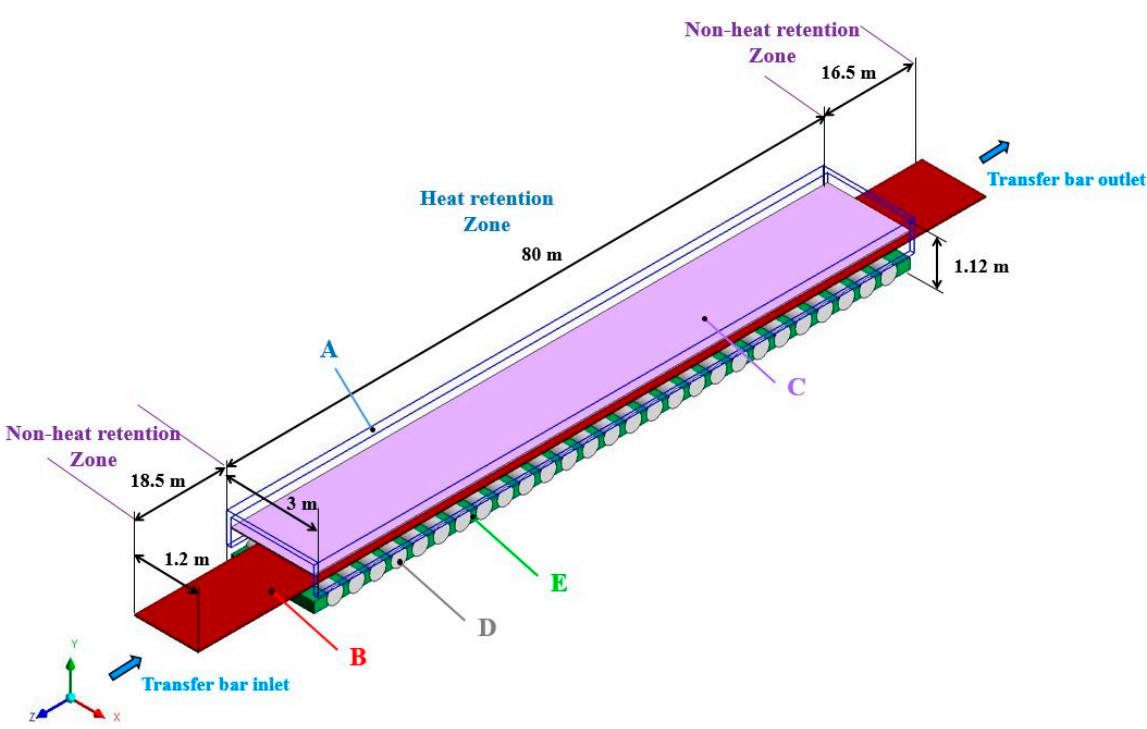

(**a**) Actual model

**Figure 3.** *Cont.*

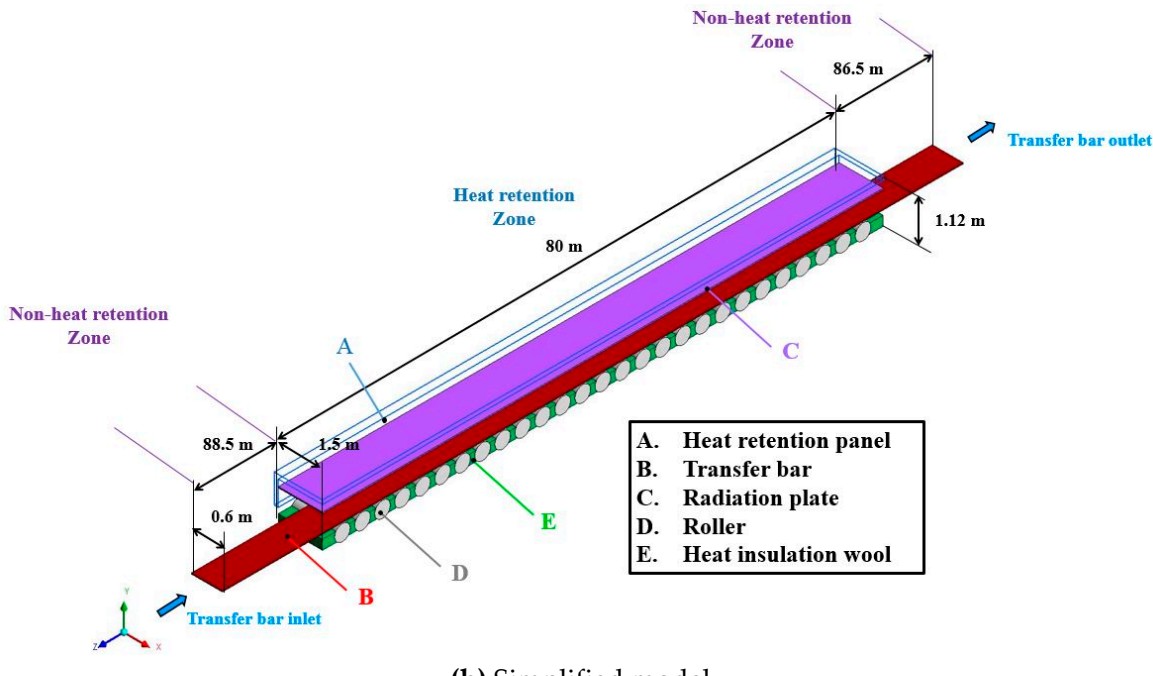

**(b)** Simplified model

**Figure 3.** Physicals models of the acting-type heat retention panel.

## 2.2. Governing Equations and Boundary Conditions

This study provides an analysis of the traditional passive heat retention panel and the acting-type heat retention panel based on the theory of fluid mechanics and the theory of heat transfer. The calculation domain is composed of a fluid domain and a solid domain. The solid domain includes the transfer bars, the heat retention panel, and the rollers beneath the conveyors. The fluid domain includes the air and the transfer bars inside the heat retention panel. The transfer bars are assumed to be a moving fluid coupled with the laminar flow passing through the heat retention panel. The continuity equation, momentum equation, and energy equation are as follows:

$$\frac{\partial \rho}{\partial t} + \frac{\partial(\rho u_j)}{\partial x_j} = 0, \tag{1}$$

$$\frac{\partial(\rho u_i)}{\partial t} + \frac{\partial(\rho u_i u_j)}{\partial x_j} = -\frac{\partial P}{\partial x_i} + \frac{\partial}{\partial x_j}\left[\mu\left(\frac{\partial u_i}{\partial x_j} + \frac{\partial u_j}{\partial x_i}\right)\right], \tag{2}$$

$$\frac{\partial T}{\partial t} + \frac{\partial(u_j T)}{\partial x_j} = \alpha(T)\frac{\partial}{\partial x_j}\left(\frac{\partial T}{\partial x_j}\right) - \nabla \cdot \vec{q}_{rad}, \tag{3}$$

where $\nabla \cdot \vec{q}_{rad}$ is the radiative heat transfer source term. The temperature field of the solid domain is governed by the following transient conduction equation:

$$\frac{\partial}{\partial x_j}k\left(\frac{\partial T}{\partial x_j}\right) + q = \rho C_p \frac{\partial T}{\partial t}, \tag{4}$$

where $\rho$, $C_p$, and k are the density, heat capacity, and the thermal conductivity of the solid domain, respectively. The convective heat transfer between the surroundings and the solid surface outside the heat retention panel is calculated using the following equation:

$$q_s^C = h(T_s - T_\infty), \tag{5}$$

where h is the convective heat transfer coefficient for the air on the solid surface. In addition, $T_s$ and $T_\infty$ are the temperature of the air and the solid surface, respectively.

Because the effect of the radiative heat transfer dominates the temperature field within the heat retention panel, the S2S (surface-to-surface) radiation model explained in [16] is adopted for the purpose of the current study. The S2S radiation model is used to calculate the radiation exchange in an enclosure comprising gray-diffuse surfaces, for which the governing equation is described as follows:

$$q_{out,k} = \varepsilon_k \sigma \left( T_k^4 - T_0^4 \right) + \rho_k q_{in,k}, \tag{6}$$

where $\varepsilon_k$ is the emissivity of the solid surface, $\sigma$ is Boltzmann's constant, and $q_{in,k}$ is the energy flux incident on the surface from the surroundings, which can be calculated using the following equation:

$$q_{in,k} = \sum_{j=1}^{N} F_{kj} q_{out,j}, \tag{7}$$

where $q_{out,j}$ is the energy flux leaving from j surface. The energy exchange between two surfaces depends in part on their size, separation distance, and orientation. These parameters are accounted for by a geometric function called view factor that can be evaluated with the equation:

$$F_{kj} = \frac{\text{diffuse energy leaving } A_k \text{ directly toward and intercepted by } A_k}{\text{total diffuse energy leaving } A_k} = \frac{1}{A_k} \int_{A_k} \int_{A_j} \frac{cos\theta_k cos\theta_j}{\pi r^2} \delta_{kj} dA_j A_k, \tag{8}$$

The word directly is meant to imply "on a straight path without intervening reflections". Additionally, the formulation of $F_{kj}$ must ensure that all surfaces are diffuse surfaces with uniform radiosity. The symmetry boundary conditions of the numerical model are expressed as follows:

$$\frac{\partial u_i}{\partial n} = 0, \tag{9}$$

$$\frac{\partial T}{\partial n} = 0, \tag{10}$$

Inside the heat retention panel, the solid surfaces and the air are conjugated heat transfer boundaries, which can be expressed as:

$$u_i \left[ \frac{\partial (\rho C_p T)}{\partial x_i} \right] = \frac{\partial}{\partial x_i} \left( k \frac{\partial T}{\partial x_i} \right), \tag{11}$$

At the transfer bar inlet of the heat retention panel model, the initial temperature of the transfer bar is assumed to be uniform at $T_{in} = 1192$ °C, and the variations in the running speed range from 6 m/s to 2 m/s. The initial temperature of the heat retention panel, which includes the traditional passive heat retention and the acting-type heat retention panel, rollers, and radiation plate, are 400 °C, 200 °C, and 1095 °C, respectively. Furthermore, in general, the heat source of the acting-type heat retention panel is supported by burning natural gas or using an electric heater to heat the upper surface of the radiation plate in general. In order to simplify the complex heating processes, the heat flux boundaries are substituted, as shown in Figure 4b, for the burning process and the electric heating process mentioned above.

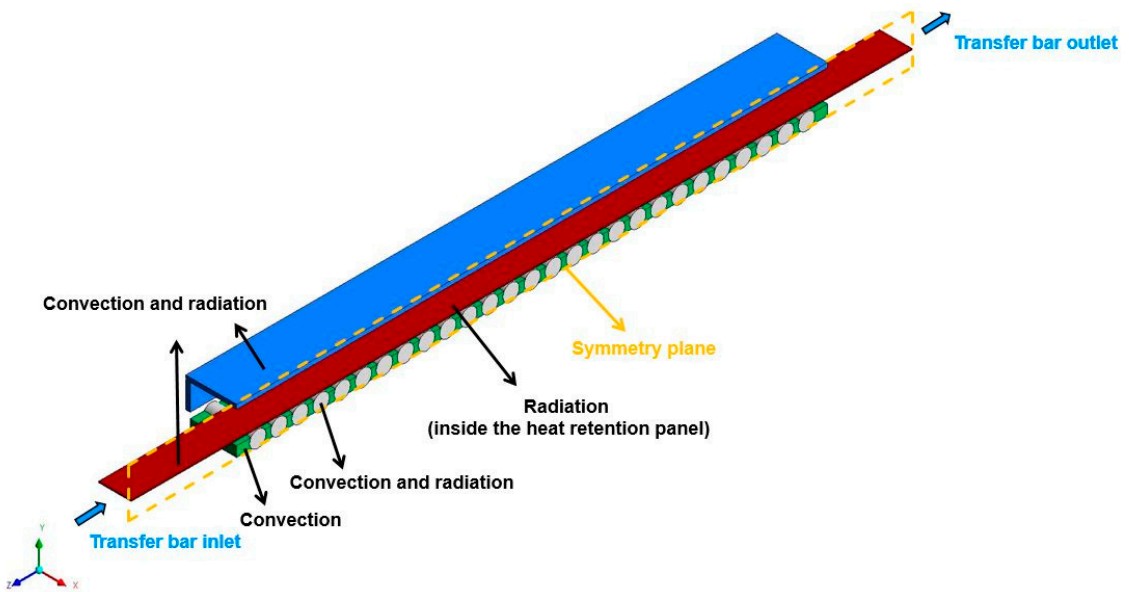

(**a**) Traditional passive heat retention panel

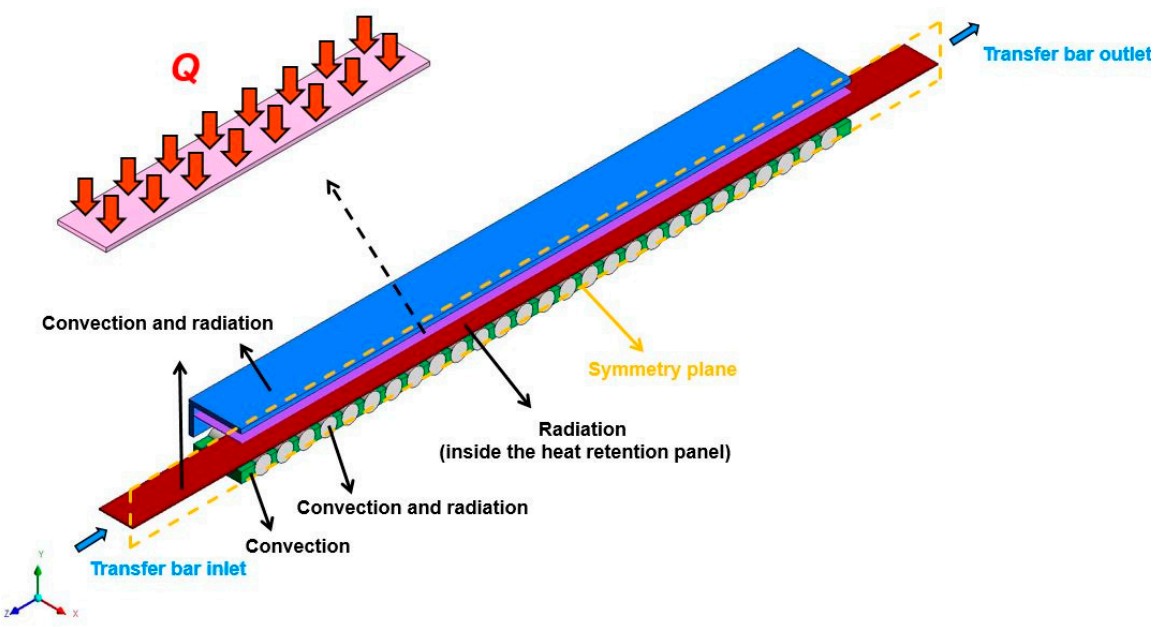

(**b**) Acting-type heat retention panel

**Figure 4.** Boundary conditions of the numerical models.

### 2.3. Numerical Methods and Grid Independence

In this study, the commercial software ANSYS-Fluent is adopted to solve the governing equations. The finite volume method (FVM) incorporates a second order upwind scheme and a first order implicit scheme for transient formulation. From CSC research, the convective heat transfer effect of air inside the panel is usually neglected, and the heat transfer is dominated by the radiation. Thus, a surface to surface (S2S) radiation model in fluent is adopted to solve the heat transfer problem. Figure 5 shows the computational grids of the three-dimensional traditional passive heat retention panel model, which is composed of 3,873,445 cells, and Figure 6 shows the computational grids of the three-dimensional acting-type heat retention panel model, which is composed of 3,445,396 cells.

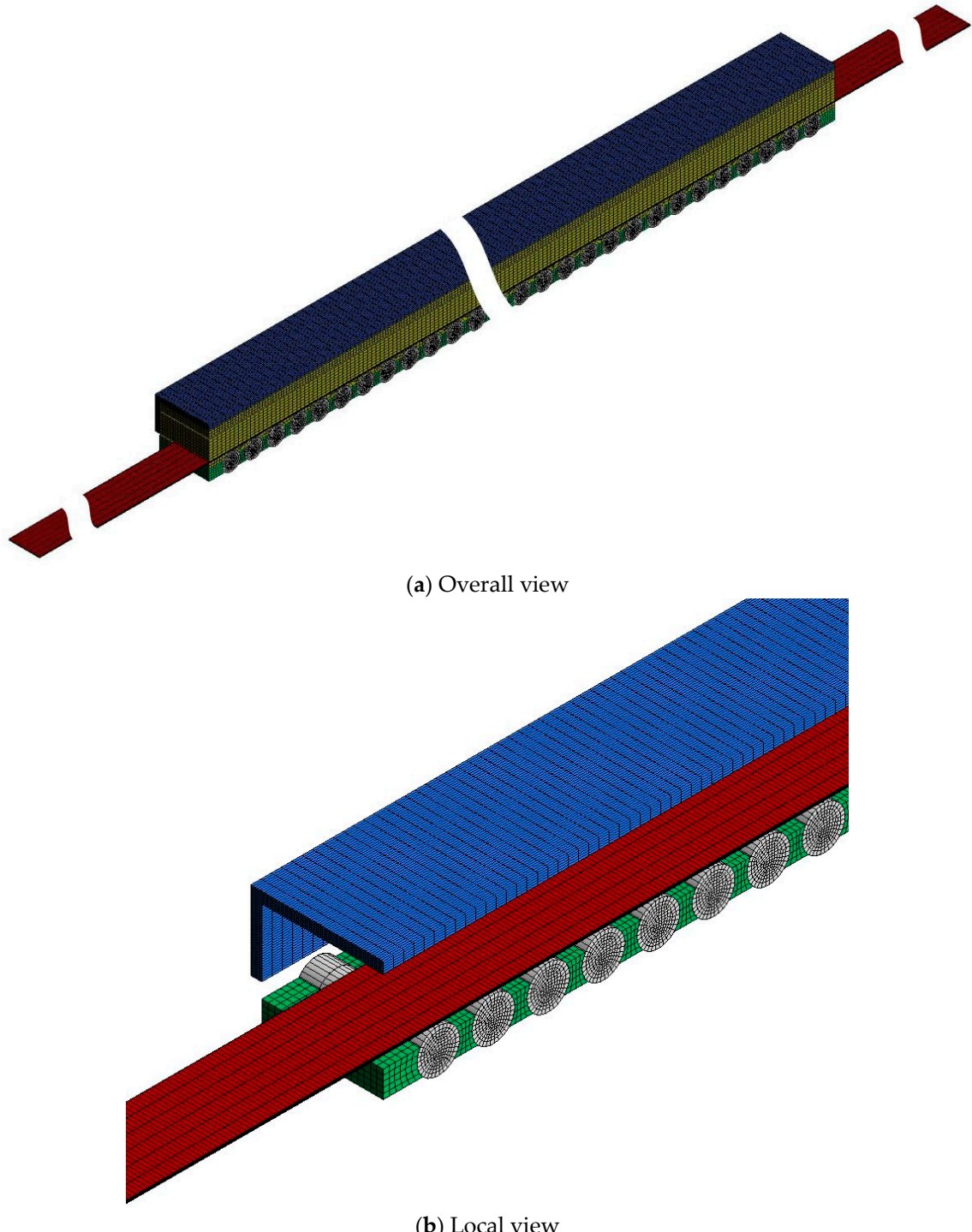

(**a**) Overall view

(**b**) Local view

**Figure 5.** Computational grid systems of the traditional passive heat retention panel.

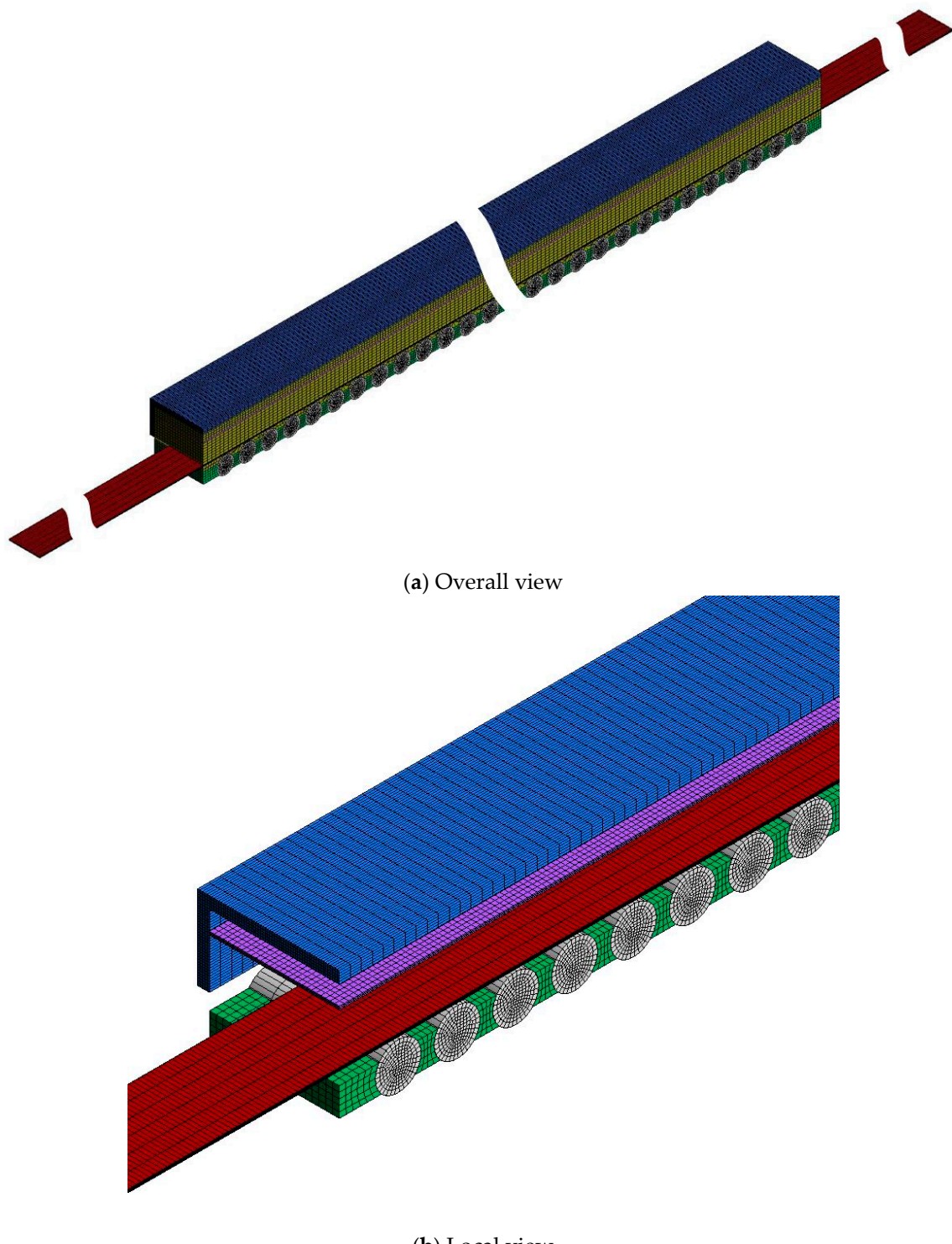

(**a**) Overall view

(**b**) Local view

**Figure 6.** Computational grid systems of the acting-type heat retention panel.

As shown in Figure 7, for the three-dimensional traditional passive heat retention panel model, four different grid numbers, 3,070,833 cells, 3,435,267 cells, 3,873,445 cells, and 4,242,534 cells, were tested for the time step of 0.5 s. The deviations in the transfer bar head for the four grid numbers were 1.22%, 1.04%, 0.91%, and 0.89%, respectively. Moreover, the deviations in the transfer bar tail for the four grid numbers were 1.31%, 1.13%, 1.06%, and 0.94%, respectively. The predicted results for the third grid and the fourth grid were almost the same. Thus, the third grid was adopted. It could also satisfy the requirements related to grid independence, computational accuracy, and

computational time needed. The discretized system was solved iteratively until it satisfied the residual convergence criterion as follow:

$$R = \left| \left( Q_c^{n+1} - Q_c^n \right) + \left( Q_d^{n+1} - Q_d^n \right) + \left( Q_s^{n+1} - Q_s^n \right) \right|, \tag{12}$$

where $Q_c$ is the value of convection term in difference equation, $Q_d$ is the value of diffusion term in difference equation, and $Q_s$ is the value of source term in difference equation. In this study, the criterion for numerical convergence judgment is to make the sum of the residual values of all grids satisfy the maximum relative error less than $10^{-3}$ and $10^{-6}$, respectively.

$$\begin{aligned} \sum \left| R_\varphi \right| &< 1 \times 10^{-3}, \; \varphi = u, v, w, P, \\ \sum \left| R_\varphi \right| &< 1 \times 10^{-6}, \; \varphi = T, \end{aligned} \tag{13}$$

The simulations were performed as a parallel calculation with sixteen core central processing units for both the traditional passive heat retention panel and the acting-type heat retention panel models. The computer calculation time was around 6 h and 7 h for the traditional passive and the acting-type models, respectively.

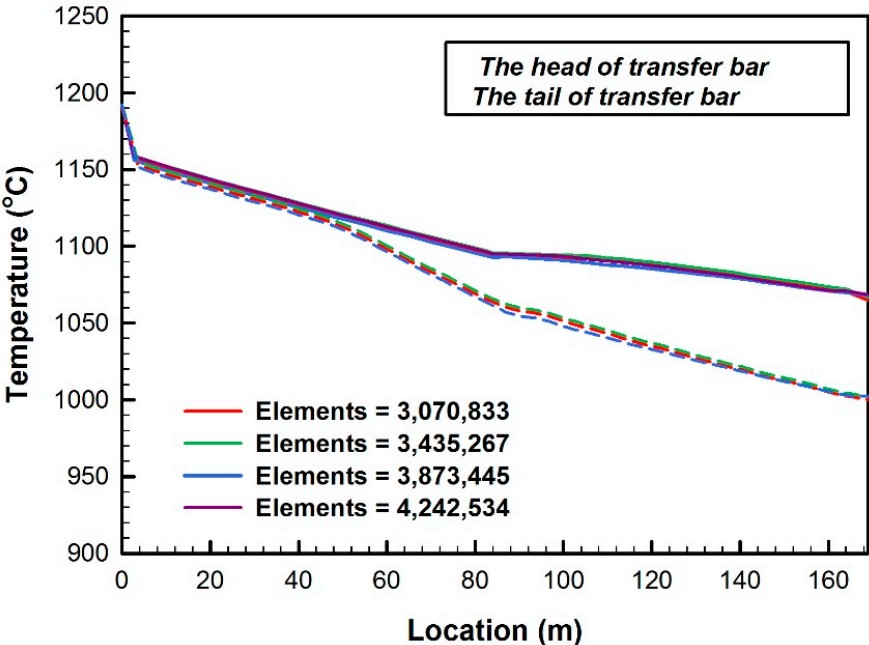

**Figure 7.** Computational grid tests for the traditional passive heat retention panel model.

## 3. Results and Discussion

### 3.1. Traditional Passive Heat Retention Panel

To validate the numerical models developed in this study, the results of the transfer bar for the traditional passive heat retention panel model were compared with the in-situ data provided by CSC, as shown in Table 2. The in-situ temperatures were obtained with R2DT and FET tests by measuring the upper surfaces of the transfer bar when it was passing through the traditional passive heat retention panel. R2DT is the abbreviation of the temperature detector near the roughing mills and FET is the abbreviation of the temperature detector near the finishing mills. Table 2 shows that the in-situ temperature of the head was 1097 °C as measured using the R2DT, and the numerical temperature was 1092.78 °C. The deviation between the in-situ and the numerical temperature was approximately 0.38%. The in-situ temperature of the head was 1078 °C as measured using the FET, and the numerical temperature was 1068.24 °C. The deviation between the in-situ and the numerical

temperature was approximately 0.91%. The in-situ temperature of the tail was 1081 °C as measured using the R2DT, and the numerical temperature was 1061.55 °C. The deviation between the in-situ and the numerical temperature was approximately 1.79%. The in-situ temperature of the tail was 1013.33 °C using the FET, and the numerical temperature was 1002.58 °C. The deviation between the in-situ and the numerical temperature was approximately 1.06%. As mentioned above, the numerical temperature difference in the head and the tail was 65.66 °C, which is close to the in-situ temperature, 64.67 °C.

**Table 2.** Comparison of the numerical data with the in-situ data for the upper surface of the transfer bar with the traditional passive heat retention panel.

|  | R2DT | | | FET | | |
|---|---|---|---|---|---|---|
|  | **Num. (°C)** | **In-Situ (°C)** | **Deviation (%)** | **Num. (°C)** | **In-Situ (°C)** | **Deviation (%)** |
| Head | 1092.78 | 1097.00 | 0.38 | 1068.24 | 1078.00 | 0.91 |
| Tail | 1061.55 | 1081.00 | 1.79 | 1002.58 | 1013.33 | 1.06 |
| ΔT | 31.23 | 16.00 | - | 65.66 | 64.67 | - |

Figure 8 displays the temperature distributions of the transfer bar of numerical results and in-situ data from CSC. After the head of the transfer bar entered the heat retention panel (at the position of 90.02 m), it slowed down to 5 m/s at 139.22 m and then slowed down again to 2 m/s at 151.37 m. The residence time inside the heat retention panel of the tail is increased as the running speed of the head is decreased. This speed difference causes a temperature drop of the tail is larger than the head of the transfer bar. Eventually, the temperature difference is generated when the head and the tail reach the FET position. Figure 9 shows the temperature contours of the traditional passive heat retention panel model. It is obvious that the temperature difference was generated after the transfer bar passed through the heat retention panel. The local temperature contours of the heat retention panel, rollers, and the insulation wool are presented in Figure 10.

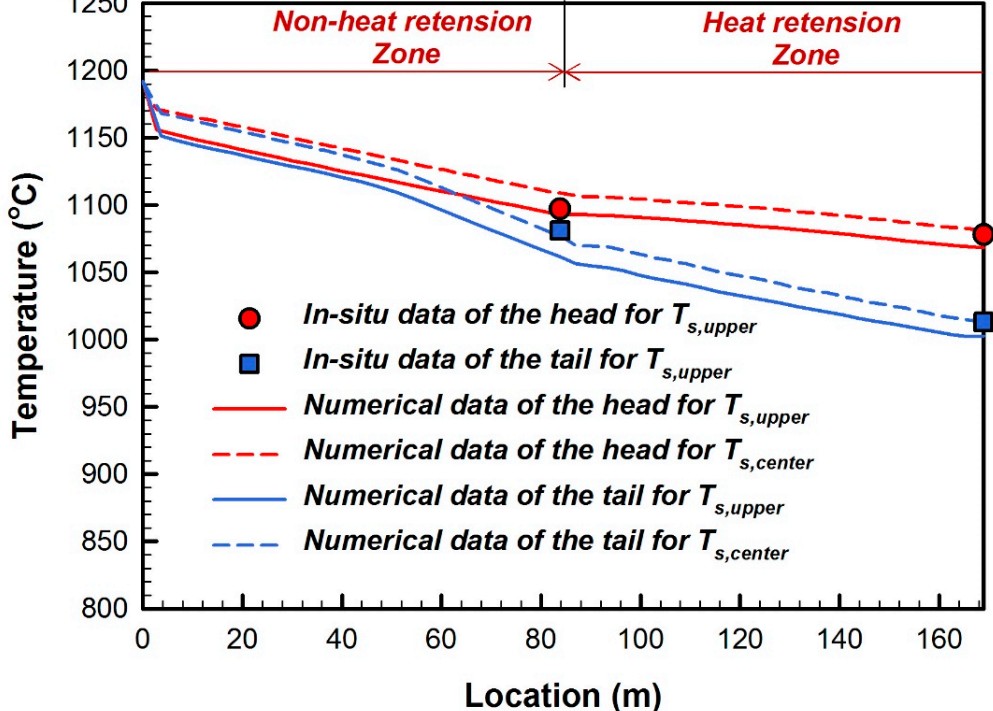

**Figure 8.** Comparison of the numerical data with the in-situ data for the upper surface and centerline-surface of the transfer bar with the traditional passive heat retention panel.

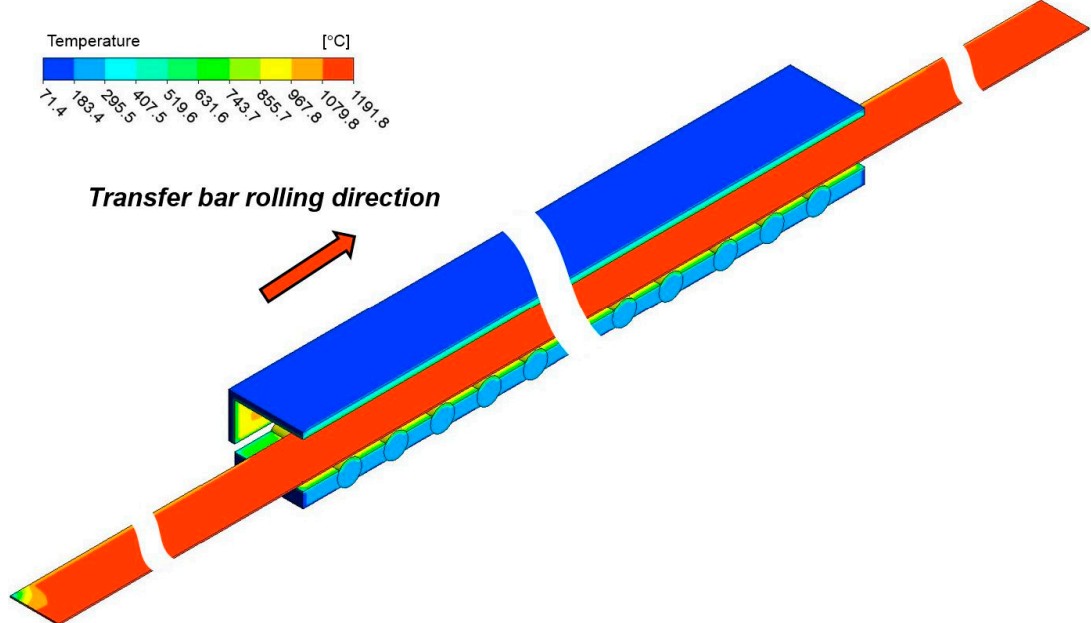

**Figure 9.** Temperature contours of the traditional passive heat retention panel model (overall view).

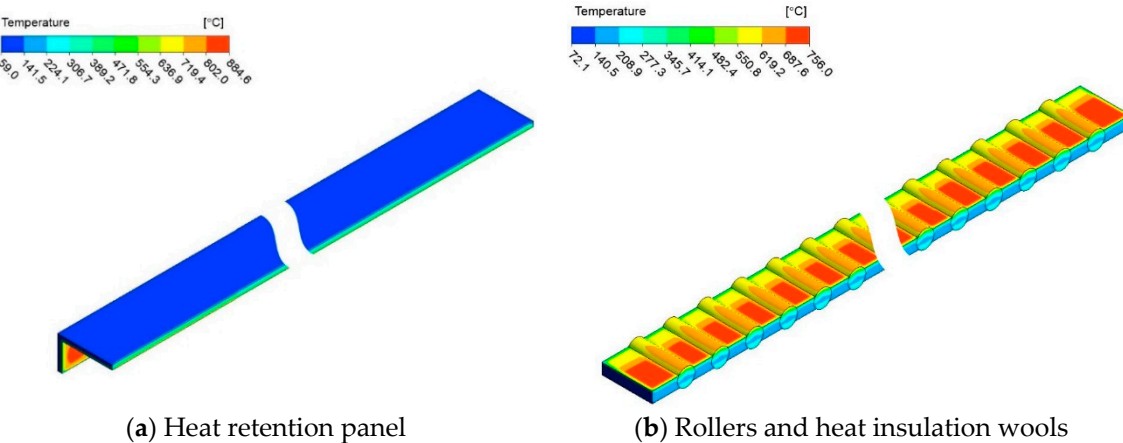

(**a**) Heat retention panel  (**b**) Rollers and heat insulation wools

**Figure 10.** Temperature contours of the traditional passive heat retention panel model (local view).

Table 3 shows that the numerical temperatures of the outer and inner wall of the heat retention panel were 114 °C and 711 °C, respectively. In addition, the numerical temperatures of the upper and lower surface of the rollers were 622 °C and 238 °C, respectively. As mentioned above, all the numerical temperatures of the heat retention panel and the rollers matched the in-situ data provided by CSC.

**Table 3.** Comparison of the numerical data and the in-situ data for different surfaces with the traditional passive heat retention panel.

|  |  | Num. (°C) | In-Situ (°C) | Deviation (%) |
|---|---|---|---|---|
| Heat retention panel | Outer surface | 114 | 100~150 | 8.80 |
|  | Inner surface | 711 | 600~750 | 5.33 |
| Rollers | Upper surface | 622 | 600~700 | 4.31 |
|  | Lower surface | 238 | 200~300 | 4.80 |

### 3.2. Acting-Type Heat Retention Panel

Based on the boundary parameters used in the traditional passive heat retention panel model, this study investigated the temperature difference in the acting-type heat retention panel that was not developed at CSC. According to the heat flux given on the upper surface of the radiation plate, the results are divided into two cases. Case 1 is given 462 kW/m$^2$ heat flux, and Case 2 is given 840 kW/m$^2$, respectively. Table 4 presents the detailed numerical temperatures for Case 1 and Case 2 of the head and the tail of the transfer bar. The numerical temperature of the head was 1096.72 °C at the R2DT position, which was quite close to the in-situ temperature of the traditional passive heat retention panel. The numerical temperature of the tail was 1065.73 °C at the R2DT position, which is matched with the in-situ temperatures of the traditional passive heat retention panel. Essentially, the temperatures for the acting-type model of the head and the tail at the R2DT position are almost the same as those for the traditional passive heat retention panel.

**Table 4.** Numerical data for the upper surface of the transfer bar with the acting-type heat retention panel.

| | R2DT | | FET | |
|---|---|---|---|---|
| Temperature (°C) | Case 1 | Case 2 | Case 1 | Case 2 |
| Head | 1096.72 | 1096.72 | 1112.73 | 1121.44 |
| Tail | 1065.73 | 1065.73 | 1081.51 | 1119.47 |
| ΔT | 30.99 | 30.99 | 31.22 | 1.97 |

The purpose in this work was to investigate whether the temperature difference between the head and the tail can be reduced after the transfer bar is heated by the radiation plate, so the finishing rolling process can go smoothly. The temperature distributions for Case 1 are shown in Figure 11, where it is obvious that the head and the tail were directly heated by the radiative heat transfer of the radiation plate. The temperature distribution of the head rose less than that of the tail due to the variations in the running speed. The final temperature difference at the FET position decreased from 65.66 °C to 31.22 °C.

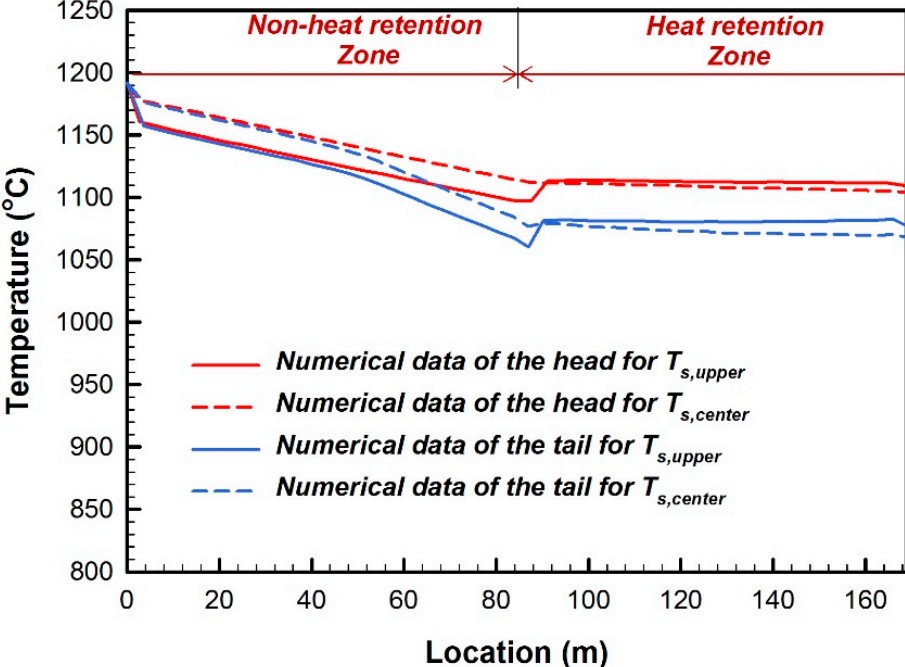

**Figure 11.** Numerical data for the upper surface and centerline-surface of the transfer bar with the acting-type heat retention panel (Case 1).

**Table 5.** Numerical data for different surfaces with the acting-type heat retention panel.

| Temperature (°C) | | Case 1 | Case 2 |
|---|---|---|---|
| Heat retention panel | Outer surface | 130.95 | 131.76 |
| | Inner surface | 916.17 | 1326.50 |
| Rollers | Upper surface | 775.31 | 799.42 |
| | Lower surface | 227.51 | 251.69 |

However, Table 5 shows that the temperature values of the inner walls inside the heat retention panel and the upper surfaces of the rollers rose to 916.17 °C and 775.31 °C for Case 1, as well as to 1326.50 °C and 799.42 °C for Case 2, respectively. Figure 12 shows the temperature contours of the acting-type heat retention panel model, and Figure 13 shows the local temperature contours of the heat retention panel, rollers, and the insulation wool. The temperature values were almost higher than the traditional passive heat retention panel model.

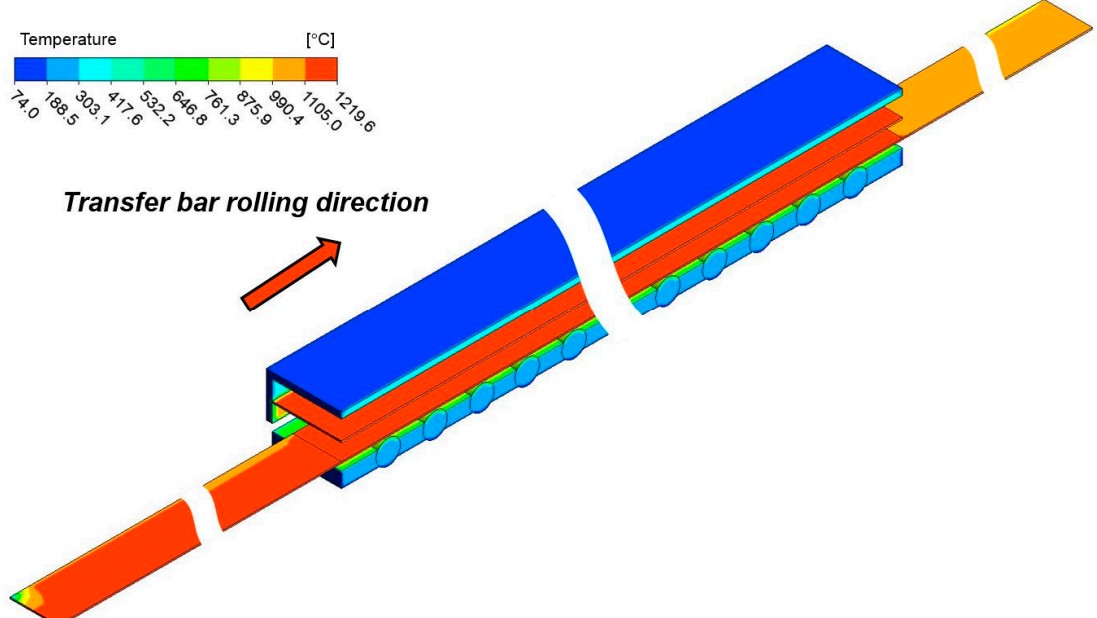

**Figure 12.** Temperature contours of the acting-type heat retention panel model (Case 1) (Overall view).

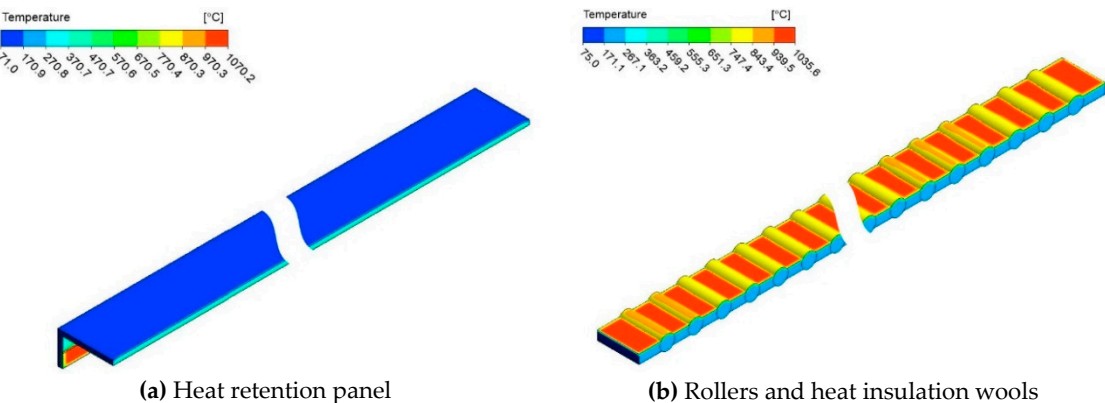

**(a)** Heat retention panel             **(b)** Rollers and heat insulation wools

**Figure 13.** Temperature contours of the acting-type heat retention panel model (Case 1) (local view).

Figure 14 shows the temperature distributions for Case 2. When the heat flux was increased to 840 kW/m$^2$, the temperature difference at the FET position was reduced to 1.97 °C. The temperature distribution of the tail rose more intensely compared with Case 1. Meanwhile, the temperature values of the inner walls inside the heat retention panel and the upper surfaces of the rollers rose to 1329.4 °C and 791.58 °C, respectively. Generally, the unit of heat used in the steel industry is kW/ton, which refers to the amount of fuel required to raise each ton of steel by one degree. In this study, the heat flux 462 kW/m$^2$ and 840 kW/m$^2$ were converted to 1743.47 kW/ton and 3169.94 kW/ton, respectively. Meanwhile, Figure 15 shows the relationship between the amount of fuel required and the temperature difference, which is helpful for steel plants to determine how much fuel will be used when using the acting-type heat retention panel.

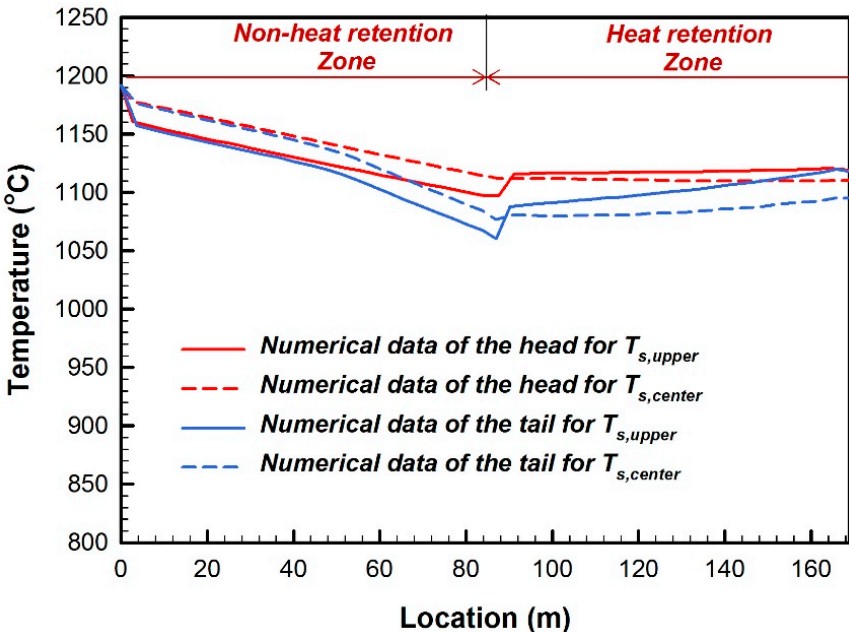

**Figure 14.** Numerical data for the upper surface and centerline-surface of the transfer bar with the acting-type heat retention panel (Case 2).

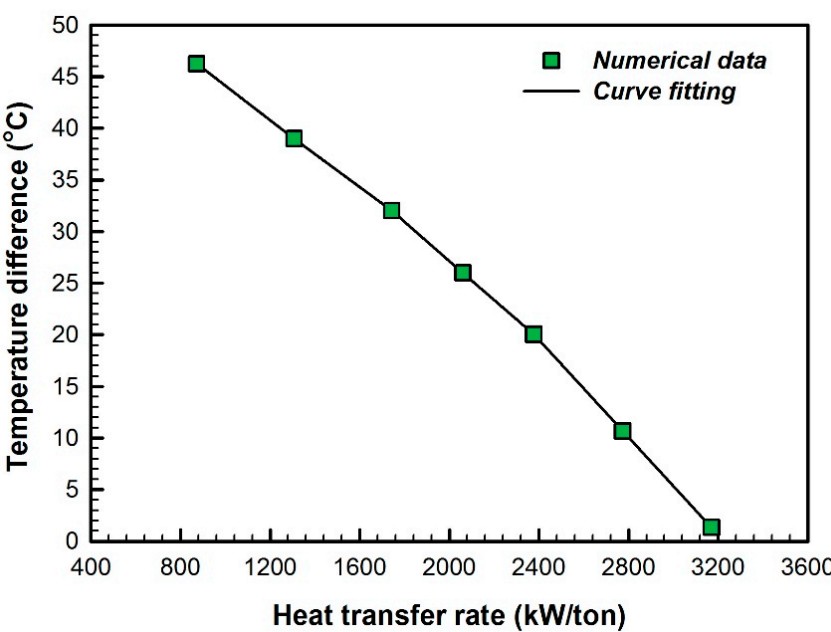

**Figure 15.** Relationship between the temperature differences in the transfer bars and the fuel used.

## 4. Conclusions

In this study, the traditional passive heat retention panel numerical model combined with UDF was adopted to simulate the temperature distributions and temperature differences when the transfer bars pass through the heat retention panel at different running speeds.

For the traditional passive heat retention panel, the deviation in the temperature difference in the transfer bar at FET position between the numerical simulation and the in-situ data was about 1.53%. Moreover, it was found that the temperature difference was induced by variations in the running speed of the transfer bar. Table 3 shows that the temperature deviations between the numerical simulation and the in-situ data for the inner and outer walls of the heat retention panel were 8.80% and 5.33%, respectively. The temperature deviations between the numerical simulation and the in-situ data of the inner and outer walls of the rollers were 4.31% and 4.80%, respectively. Since the performance of the numerical results worked well compared with the in-situ data, the corresponding parameters, including the initial temperature, convective heat transfer coefficient of the surroundings, and emissivity of the solid surfaces, could be used in the acting-type heat retention panel numerical model.

According to the numerical results of the acting-type heat retention panel model, providing the two heat fluxes on the upper surface of the radiation plate is an effective method by which to replace the burning process or the electric heating process. The numerical simulation indicates that the transfer bars can be heated by the radiation plate. When the heat flux increases, the temperature difference between the head and the tail of the transfer bar will be reduced. In contrast, the heat flux will cause the temperatures of the heat retention panel and the rollers to increase. Eventually, by converting the heat flux to the amount of fuel required, steel plants can obtain the relevant information about their energy consumption.

**Author Contributions:** All authors contributed to this work. J.-Y.J. and C.-C.C. performed the theoretical model. J.-W.G. executed the numerical simulation work.

**Funding:** This research was funded by the China Steel Corporation, Taiwan.

**Acknowledgments:** The financial support by China Steel Corporation, Taiwan is highly appreciated.

**Conflicts of Interest:** The authors declare no conflict of interest.

## Nomenclature

| | |
|---|---|
| $A$ | area ($m^2$) |
| $a$ | acceleration ($m/s^2$) |
| $C_p$ | specific heat (kJ/kg·K) |
| $F_{kj}$ | view factor from $k$ surface to $j$ surface |
| $h$ | convective heat transfer coefficient ($W/m^2$·K) |
| $k$ | thermal conductivity (W/m·K) |
| $n$ | normal direction |
| $P$ | pressure (Pa) |
| $q$ | heat flux ($W/m^2$) |
| $R$ | numerical residual |
| $T$ | temperature (°C) |
| $t$ | time (s) |
| $u,v,w$ | velocity (m/s) |
| $x,y,z$ | coordinates |
| **Greek symbols** | |
| $\alpha$ | thermal diffusivity ($m^2/s$) |
| $\delta_{kj}$ | Kronecker delta |
| $\varepsilon$ | emissivity |
| $\varphi$ | property of fluid |
| $\mu$ | viscosity (N s/$m^2$) |

| | |
|---|---|
| $\rho$ | density (kg/m$^3$); reflectance |
| $\sigma$ | Boltzmann's constant (J/K) |
| $\sum$ | summation |

**Subscripts**

| | |
|---|---|
| *in* | state of inlet |
| $\infty$ | surroundings |
| *out* | state of outlet |
| *s* | surface |

**Superscripts**

| | |
|---|---|
| *C* | convection |

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
