# Peer review of "The Heat Transfer Analysis of an Acting-type Heat Retention Panel used in a Hot Rolling Process"

_applsci, doi:10.3390/app9010189_

Round 1
Reviewer 1 Report
The scientific paper presents an application of a numerical 3D thermal analysis concerning an industrial hot strip rolling process in order to analysis and minimizes the temperature difference between the head and the tail of the transfer bars. Effects of a panel heat radiation are taken into account using original convolution of the surface thermal radiation flux.
The authors have been structured the paper in four principal parts: Introduction, Mathematical Analysis, Results and Discussions. After a short synthesis of actual industrial goals and recent used modelling approaches to simulate temperature distribution and evolution during a rolling process, a physical description of a specific China Steel Corporation rolling is detailed together with the specific design, motion and thermal governing equations used by ANSYS – Fluent commercial code. The main objective is to improve the quality of steel rolling products by estimation and decrease of the temperature difference between the head and the tail of the transfer bar. The obtained numerical results are firstly validated by comparisons with in-situ temperature measurements on two specific locations. Starting from a lot of simulations, authors plot the variation of the temperature differences with the heat transfer rate using the acting-type heat holding panel. It is concluded than when the heat flux increases, the temperature difference between the head and the tail of the transfer bar will be reduced.
However a lot of minor points must be revised and clarified by authors:
1° Page 1 and Page 2 – Introduction: if possibility to add more references linked to numerical finite element simulation of 3D rolling processes.
2° Page 6 – Line 131: is is necessary to precise the expression of qrad vector.
3° Page 6 – Line 142 – Eq. (6): the emissivity term must be replaced by epsilon*sigma* (T^4-T0^4). It is required to verify if for the presented numerical results this real form of emissivity term has a major impact as compared to the used no-physical form epsilon*sigma*T^4.
4° Page 6 – Line 145 – Eq. (7): the authors must explicit the qout,j term.
5° Page 6 – Line 148 – Eq. (8): details of Fkj computation together with the significance of teta_k and teta_j, Aj and Ak terms.
6° Page 11 – Line 183 – Eq. (12): is is necessary to explain if residual R_fi term is computed in an absolute or relative form. Contrary the R_fi is a dimensional term and the numerical tolerance can be insufficient to obtain a true numerical convergency.
Regarding the proposed original numerical tri-dimensional study concerning influence of the thermal panel radiation flux used during a hot rolling process, if authors take into account all the above mentioned remarks and compulsory proposed minor corrections, the paper can be accepted for publication in Applied Sciences – MDPI Journal.
Reviewer 2 Report
This manuscript analyzes an industrial process with the help of a popular CFD software. The merits of the process are very intriguing from the engineering perspective, however, I think that some fundamental aspects of the model are lacking and the results are hastily presented. I cannot recommend publication in the current form, until the following issues are addressed:
The term "heat holding" is somewhat awkward to me. I don't believe that it is a standard technical term -- as the matter of fact, the included references use the term "temperature holding" instead -- and therefore, the authors should consider another more elegant term.
The authors mention in the abstract the use of user-defined functions. However, they don't elaborate, not even slightly in the rest of the manuscript. I suggest removing it from the abstract.
I am a little bit confused about the inclusion of air convection. Do the authors solve for the velocity/temperature field of air or do they just use Newton's cooling law at the walls? If the former is true, why don't they present the velocity field and the contribution of convection to the overall heat transfer? The authors should clarify.
The authors should back up theoretically and/or computationally the choice of the 1/3 symmetry domain. I don't feel that this is a valid choice, but only a 1/2 symmetry is acceptable. Could the authors compare the 2 symmetries. I don't believe a 1/2 symmetry should add up to the overall computational cost considerably. Besides the authors report a few hours of computational time, which is quite low.
The number of references is unacceptably small (9 references only!). In fact, I believe that only 3 of them are actually from peer reviewed journals. The authors should add more.
Round 2
Reviewer 2 Report
The authors addressed all of my concerns, except for the one with the symmetry conditions. In fact they seemed to dismiss it entirely, by only citing the computational times for the 1/3 and 1/2 domains.
I don't believe that an additional simulation that will last 8 hours (as the authors claim) with the 1/2 domain is too much to ask. And besides, this will strengthen the authors' work. Unless, the use of a 1/2 domain produces different results, in which case they should perform the simulations again and rewrite the manuscript.
I still do not recommend publication until my concern is properly addressed.
Round 3
